# Blood lead levels and lead toxicity in children aged 1-5 years of Cinangka Village, Bogor Regency

Yana Irawati[1,2]*, Haryoto Kusnoputranto[3], Umar Fahmi Achmadi[3], Ahmad Safrudin[4], Alfred Sitorus[4], Rifqi Risandi[5], Suradi Wangsamuda[5], Puji Budi Setia Asih[5], Din Syafruddin[5,6]

1 Faculty of Public Health, Universitas Indonesia, Depok, Indonesia, 2 Balai Besar Pelatihan Kesehatan Jakarta, Ministry of Health, Jakarta, Indonesia, 3 Environmental Health Department, Faculty of Public Health, Universitas Indonesia, Depok, Indonesia, 4 Komite Penghapusan Bensin Bertimbal, Jakarta, Indonesia, 5 Eijkman Institute for Molecular Biology, Jakarta, Indonesia, 6 Hasanuddin University Medical Research Unit (HUMRC), Makassar, Indonesia

* mllentu@gmail.com

**Data Availability Statement:** All relevant data are within the paper and its Supporting Information files.

## Abstract

Lead is one of ten hazardous chemicals of public health concern and is used in more than 900 occupations, including the battery, smelting, and mining industries. Lead toxicity accounts for 1.5% (900,000) of deaths annually in the world. In Indonesia, reports of high Blood Lead Level (BLL) were associated with residency in Used Lead Acid Battery (ULAB) recycling sites. The present study aims to investigate the BLL and the evidence of lead toxicity of children living in an ULAB recycling site in Bogor Regency, Indonesia. A cross-sectional study involving 128 children aged 1–5 years was conducted in September-October 2019. The socio-economic factors, BLL, nutritional status, and hematological parameters, were evaluated. Data were analyzed by univariate and bivariate using the Chi-Square test. Socio-economic factors revealed only 2.3% children have pica and 10.9% children have hand-to-mouth habits. Majority of parents had low income, education, and have stayed in the village for years. Analysis on BLL revealed that 69.5% children had BLL of >10 μg/dL, 25% had abnormal BMI, 23.4% had underweight, 53.9% had stunting, 33.6% had anemia, and 22.6% had basophilic stippling. The average BLL and hemoglobin levels of respondents were 17.03 μg/dL and 11.48 g/dL, respectively. Bivariate analysis revealed that children with high BLL had double risk of having underweight and protected from stunting. Analysis on the association between BLL and BMI for age revealed a higher risk to have abnormal BMI. The high BLL also had 1.017 times risk of developing anemia, and almost doubled risk of having basophilic stippling, although they were not statistically significant. In conclusion, the high BLL of children living in the ULAB recycling indicates that lead exposure as well as lead toxicity are still occurring in Cinangka Village, and alerts to the need for a systematic action to mitigate the exposure.

**Funding:** This research funding was supported Ministry of Research and Technology/National Agency for Research and Innovation (RISTEK-BRIN), through Eijkman Institute for Molecular Biology, and Indonesia Endowment Fund for Education, abbreviated LPDP (Lembaga Pengelola Dana Pendidikan), Ministry of Finance, Indonesia (PRJ-26/LPDP.4/2020) Yes. Eijkman Institute for Molecular Biology support in the field assistances, laboratory consumables and equipment facilities during the research. Also had role in data collection and analysis, decision to publish, or preparation of the manuscript. No, LPDP had no rule in study design, data collection and analysis, decision to publish, or preparation of the manuscript.

**Competing interests:** The authors have declared that no competing interests exist.

**Abbreviations:** BLL, Blood Lead Level; BMI, Body Mass Index; ULAB, Used Lead Acid Batteries; OR, Odd Ratio; ARI, Acute Respiratory Infection; TCLP, Toxicity Characteristic Leaching Procedure; WHO, World Health Organization; HIV, Human Immunodeficiency Virus; AIDS, Acquired Immune Deficiency Syndrome; IHME, Institute for Health Metrics and Evaluation; DALYs, Disability-Adjusted Life Years; IQ, Intellectual Quotient; US EPA, United States Environmental Protection Agency; US CDC, United States Centers for Disease Control and Prevention; PHC, Primary Health Center; GNP, Gross National Product.

## Introduction

Lead is one of ten hazardous chemicals of public health concern [1] and is used in more than 900 occupations, including the battery, smelting and mining industries [2]. Lead accounts for 1.5% (900,000) of deaths annually in the world, a number that is almost equivalent to the number of deaths from HIV/AIDS (954,000) and is greater than the other causes of death [3]. In 2019, the Institute for Health Metrics and Evaluation (IHME) recorded more than 902,000 deaths and 21.7 million Disability-Adjusted Life Years (DALYs) worldwide due to lead exposure [4].

Lead is persistent in the environment. That is why the use of lead in the past continues to contribute to lead accumulation on the soil surface [5]. Lead poisoning is cumulative and affects many organ systems, including the neurological, hematological, gastrointestinal, cardiovascular and renal systems [6]. Children are more susceptible to lead exposure because of their habit of putting their hands in their mouths. Children also absorb more lead, and experience the effects of lead poisoning earlier than adults, even at low levels [5].

Lead can interfere with the hematological system by inhibiting the synthesis of heme in the blood and causing anemia. Lead poisoning is often followed by a deficiency of the enzyme Pyrimidine 5'-Nucleotidase (P5'N) associated with chronic hemolysis with findings of basophilic stippling (purple-blue spots) on peripheral blood smear examination using a microscope or with the presence of intra-erythrocyte pyrimidine-containing nucleotide accumulation [7, 8].

Low BLL have an impact not just on children's cognitive performance, but also on their physical growth [9]. Low-level lead exposure seldom generates a specific disease or pathological lesion, although it does contribute to organ function loss. Lead's impacts can occur as a result of two processes: (1) effects on the endocrine organs that synthesize or produce hormones that regulate bone formation; and (2) changed bone cell functions, such as cell division, enzyme activity, or calcium messenger system dysfunction [10].

Many previous studies examined the relationship between BLL and child growth. Schwartz et al. examined 2,700 children under the age of 7 years in the National Health and Nutrition Examination Survey (NHANES) II data survey in America and found a reciprocal relationship between BLL and height in the range of 5–35 g/dL. The study concluded that exposure to low concentrations of lead impairs child growth [11]. Another study found a negative relationship between growth parameters and BLL in Greek children aged 6–9 years. An increase in lead levels of 10 g/dL was associated with a decrease in height, head circumference, and chest circumference, each of 0.86, 0.33, and 0.40 cm [12]. Study conducted on children aged 6–8 years in Mexico found that children with high BLL had shorter heights and the incidence of stunting was equal in the age group of two and three years, 29%, respectively [13]. Furthermore, children with low body weight (less nutrition) were found in the three-year age group (30%). Study involving 108 children aged 5–13 years in Korea found a significant relationship between height and BLL in children, wherein high BLL tends to decrease in height at low lead concentrations [9].

Children's BLL of more than 10 µg/dL are known to be significantly associated with the incidence of anemia [5, 14], reduce iron absorption ability and affect other hematological parameters. High BLL were associated with low serum iron and ferritin [15].

More than 200 Used Lead-Acid Batterys (ULABs) recycling sites have been identified in Indonesia, including Cinangka Village. where recycling has been done since 1978. ULAB recycling in Cinangka is carried out in a very simple way by disassembling, pouring the acid liquid directly into the river or land around settlements, and burning ULABs to get lead called 'ingots' in people's home backyards. Several battery dealers who have more capital have started to build stoves. Based on information from local government officials, the initial operation of ULABs recycling

began with 20 stoves called 'Hawu' which employs 100 workers. This number continued to increase along with the increasing demand for the lead until, in 1998, the number of stoves had reached 43 units. This illegal anthropogenic activity began to receive protests from residents because the emissions (smoke and a pungent odor) impact on the health of residents, such as Acute Respiratory Illness (ARI), kidney disorders, motoric and developmental disorders, and mental retardation. The ULABs recycling has been closed several times by the authorities, but the impetus for economic needs often encourages business actors to return to smelting clandestinely. In October 2018, based on reports from residents, the National Defense Council (Dewan Pertahanan Nasional) visited Cinangka Village to reaffirm the ban on ULAB recycling along with providing education about the impact of lead on the health of residents, especially children. This study investigates the current status of BLL in children in Cinangka Village after the closure of illegal anthropogenic activities in 2018 as well as to analyze the relationship between BLL with the socio-economic factors of the households, nutritional status, and hematological disorders.

## Materials and methods

This cross-sectional study was conducted in Cinangka Village, Bogor Regency, West Java Province, Indonesia, during September-October 2019. The location of the study site is depicted in the map (Fig 1). Cinangka Village is a rural area, located approximately 60 km from Jakarta. Total population of Cinangka is 13,252, consisted of 4,195 households with the population density of 3,898 people per km$^2$. The average income of the household is 2 million rupiah (US$ 139.28), which is far below the Gross National Product (GNP) per capita US$ 3,048 [16]. Most of the resident work as subsistence farmer and ULAB recycling. Historically, this village has been used as informal battery recycling since 1978 but was temporary closed during 2003–2004, although illegal ULAB recycling activity still ongoing till now. The study protocol was reviewed and approved by the ethical committee of Public Health, Universitas Indonesia (404/UN2.F10/PPM.00.02/2019).

The study population included children aged 1–5 years who were living in four hamlets where ULAB recycling used to occur. We ask for the assistance of village health volunteers (Posyandu cadres) to select respondents with the following inclusion criteria: children aged

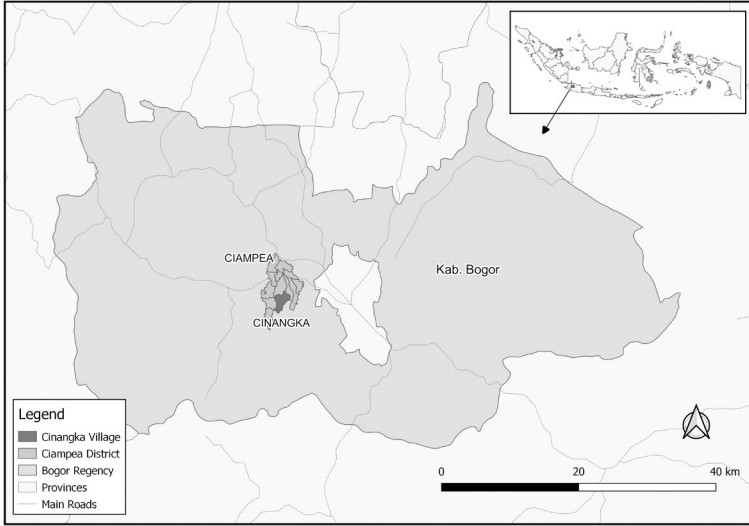

**Fig 1. Sketch's map depicting the location of the study site (Cinangka Village) in Bogor Regency and the Indonesia archipelago (https://www.naturalearthdata.com/).**

1–5 years, whose parents have lived in Cinangka Village for at least 3 years and are willing to have their children become respondents by signing informed consent. A total of 128 children were invited to participate in the study.

Socio-economic factors were evaluated through interviews with parents of the children includes behavior of the children (pica and hand to mouth), family income (in Rupiah), family expenses, parent's education, parent's length of stay, distance of the house from the ULAB former activities, drinking water source, and chip's paints. Data were analyzed by univariate and bivariate. Univariate involved a frequency distribution test. The bivariate analysis used a chi-square test with a significance level (p-value = 0.05) and 95% CI.

Nutritional status was determined by measuring children's height, body weight and Body Mass Index (BMI), based on their age, which was then entered into the WHO Anthro software to determine nutritional status in the form of abnormal BMI, underweight and stunting. The children's bodyweights and body height were measured by trained village health volunteers under the supervision of nurse at Primary Health Center (PHC), using a bathroom scale (Oxone OX 917–3, Indonesia), and height were measured using a standard mechanical stadiometer (OneMed Statue Meter, Sidoarjo, Indonesia). Body mass index was then calculated using body weight per age and height per age using the WHO Anthro program.

Blood sampling was carried out at the village health post (Posyandu) and the health volunteers arrange for the arrival of the children and their mother to have their blood drawn. A phlebotomist conducted peripheral blood sampling for this study. Each child's arm was cleaned with soap and water, dried with tissue, wiped with an alcohol swab, and dried with gauze before the procedure. Blood samples were taken through aseptic finger prick and directly pipetted it into a 0.5 ml EDTA microtainer tubes. The collected blood was used to measure BLL using LeadCare® II Portable Analyzer (ESA Biosciences, Inc., Chelmsford, USA), hemoglobin level using portable HemoCue (HemoCue AB Hb201+, Angelholm, Sweden), and to make thin blood smear for basophilic stippling, stained with Giemsa and identified under light microscopy (Objective 1000x) and dried blood spot on filter paper for DNA analysis. The reference cut off value for Hb levels is 11 g/dL [17]. Subjects who had Hb level below 11 g/dL is considered anaemic. The reference cut off value of BLL is 10 μg/dL [18]. Subjects who had BLL value above 10 μg/dL is considered high whereas as below is normal.

## Results

The subjects baseline characteristics and socio-economic factors that may relate to lead exposure is shown in Table 1. This included the average age of children, their behavior (pica and hand to mouth), family income, family expenses, parent's education, parent's length of stay, distance of the house from the ULAB former activities, drinking water source, and chipped paints. Of the 128 children enrolled, the average age was 2.79 years. Most of them were female (52.3%). Respondents came from the 4 hamlets used to be used as ULAB recycling area. Only 3 (2.3%) children have pica, and 14 (10.9%) children have hand-to-mouth habits. 77 (60.2%) of parents earned less than 2 million rupiah (US$ 139.28) per month and spent less than 1 million rupiah (US$ 69.9) per month. The majority of mothers (89.1%) had only primary education. Approximately 79.7% of parents have smoking habits, 50.0% of children lived at the ULAB recycling site for more than 23 years, while 6 (4.7%) children lived less than 50 meters from the ULAB recycling site. Children who drank well water (18.0%), and children who lived in homes with chipped paint (28.9%).

BLL of the children ranged from 4–65 μg/dL, with 69.5% had BLL of >10 μg/dL. The average value of BLL was 17.03 μg/dL (Fig 2), exceeding the US CDC threshold (10 μg/dL) [18]. Based on age, the majority of high BLL occurred in the age of 2–3 years old (32%).

**Table 1. Socio-economic characteristic of the study subjects (n = 128).**

| Variable | Total | % | Mean | Median | SD | Min-Max | 96%CI |
|---|---|---|---|---|---|---|---|
| **Ages (years)** | | | 2.79 | 2.7 | 1.13 | 1.0–5.0 | 2.59–2.99 |
| **Sex** | | | | | | | |
| Males | 61 | 47.7 | | | | | |
| Females | 67 | 52.3 | | | | | |
| **Pica** | | | | | | | |
| Yes | 3 | 2.3 | | | | | |
| No | 125 | 97.7 | | | | | |
| **Hand to mouth** | | | | | | | |
| Yes | 14 | 10.9 | | | | | |
| No | 114 | 89.1 | | | | | |
| **Family Income (Rupiahs)** | | | | | | | |
| ≤2.000.000 | 77 | 60.2 | | | | | |
| >2.000.000 | 51 | 39.8 | | | | | |
| **Family Expenses (Rupiah)** | | | | | | | |
| <1.000.000 | 60 | 46.9 | | | | | |
| ≥1.000.000 | 68 | 53.1 | | | | | |
| **Parent's Education** | | | | | | | |
| Basic | 114 | 89.1 | | | | | |
| Middle | 14 | 10.9 | | | | | |
| **Parent's Smoking Status** | | | | | | | |
| Yes | 102 | 79.7 | | | | | |
| No | 26 | 20.3 | | | | | |
| **Parent's Length of Stay (years)** | | | 22.2 | 23.5 | 11.41 | 3.0–45.0 | 20.21–24.20 |
| ≥23 years | 64 | 50 | | | | | |
| <23 years | 64 | 50 | | | | | |
| **Distance from ULAB former** | | | | | | | |
| ≤50 m | 6 | 4.7 | | | | | |
| >50 m | 112 | 95.3 | | | | | |
| **Water's source** | | | | | | | |
| Well Water | 23 | 18.0 | | | | | |
| Tap Water | 105 | 82.0 | | | | | |
| **Chip's paint** | | | | | | | |
| Yes | 37 | 28.9 | | | | | |
| No | 91 | 71.1 | | | | | |
| **Anthropometric** | | | | | | | |
| Height (cm) | | | 84.45 | 85.5 | 11.35 | 9.3–105.9 | 82.47–86.44 |
| Weight (kg) | | | 11.61 | 11.45 | 2.46 | 7.0–20.0 | 11.18–12.04 |
| **Weight for Age** | | | | | | | |
| Underweight | 30 | 23.4 | | | | | |
| Normal | 98 | 76.6 | | | | | |
| **Height for Age** | | | | | | | |
| Stunting | 69 | 53.9 | | | | | |
| Normal | 59 | 46.1 | | | | | |
| **BMI for Age** | | | | | | | |
| Abnormal | 32 | 25 | | | | | |
| Normal | 96 | 75 | | | | | |
| **Haematological parameter** | | | | | | | |

*(Continued)*

**Table 1.** (Continued)

| Variable | Total | % | Mean | Median | SD | Min-Max | 96%CI |
|---|---|---|---|---|---|---|---|
| Hb level (g/dL) | | | 11.48 | 11.5 | 1.08 | 8.9–13.9 | 11.29–11.67 |
| High | 43 | 33.6 | | | | | |
| Low | 85 | 66.4 | | | | | |
| Blood Lead Level (µg/dL) | | | 17.03 | 12.55 | 11.78 | 4.0–65.0 | 14.97–19.09 |
| High | 89 | 69.5 | | | | | |
| Low | 39 | 30.5 | | | | | |
| Basophilic stippling | | | | | | | |
| Yes | 34 | 26.6 | | | | | |
| No | 94 | 73.4 | | | | | |

Measurement of bodyweight and height of respondents revealed that 23.4% were underweight and 53.9% were stunting. In addition, BMI for age analysis revealed that 25% of the children has abnormal value in the form of wasting (4.68%) and overweight (20.32%). Bivariate analysis of the relationship between BLL and underweight indicated no significant association, but children with high BLL had a doubled risk to become underweight (OR = 2.031, p = 0,231) and protected from stunting (OR = 0.865, p = 0,854). Bivariate analysis on the association between BLL and BMI for age revealed a higher risk to have abnormal BMI (OR = 1.516, p = 1,000) (Table 2).

The average value of hemoglobin level was 11.48 g/dL (8.9–13.9 g/dL), and 33.6% of the enrolled children were anemic. The cut off for determining the status of hemoglobin level according to WHO, Hb ≥11 g/dL (normal) and Hb < 11 g/dL (anemia) [17]. Light microscopic examination of the blood smear revealed that 34 (26.6%) had basophilic stippling in their erythrocytes (Table 1).

Bivariate analysis of the BLL and hemoglobin levels showed no significant relationship between the two parameters (Table 3). Children with high BLL had a doubled risk of impaired heme synthesis, as evidenced by the presence of basophilic stippling in the erythrocytes (Fig 3), although the relationship is not statistically significant.

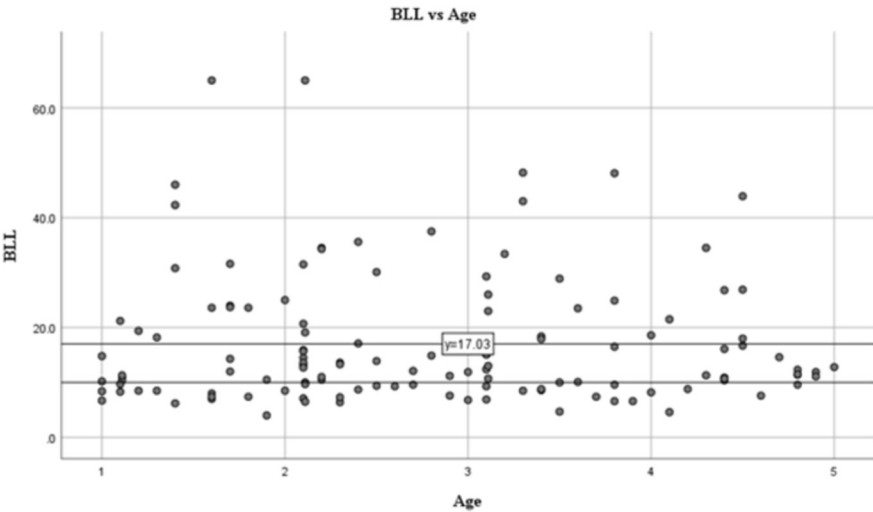

**Fig 2. Blood lead level distribution among the subjects based on age group.**

**Table 2. Relationship of BLL with nutritional status (weight for age, height for age and BMI for age).**

| BLL | Weight for Age | | | | Total | | | OR | p-value |
|---|---|---|---|---|---|---|---|---|---|
| | Underweight | | Normal | | | | | (95% CI) | |
| | n | % | n | % | n | % | | | |
| High | 24 | 27.0 | 65 | 73.0 | 89 | 100,0 | | 2.031 | 0.231 |
| Low | 6 | 15.4 | 33 | 84.6 | 39 | 100,0 | | (0.756–5.453) | |
| BLL | Height for Age | | | | Total | | | OR | p-value |
| | Stunting | | Normal | | | | | (95% CI) | |
| | n | % | n | % | n | % | | | |
| High | 47 | 52.8 | 42 | 47.2 | 89 | 100.0 | | 0.865 | 0.854 |
| Low | 22 | 56.4 | 17 | 43.5 | 39 | 100.0 | | (0.405–1.844) | |
| BLL | BMI for Age | | | | Total | | | OR | p-value |
| | Abnormal | | Normal | | | | | (95% CI) | |
| | n | % | n | % | n | % | | | |
| High | 31 | 24.8 | 94 | 75.2 | 125 | 100,0 | | 1.516 | 1.000 |
| Low | 1 | 33.3 | 2 | 66.7 | 3 | 100,0 | | (0.133–17.300) | |

## Discussion

Among the socio-economic characteristics identified, it is evident that the majority of the parents of respondent in the study site have a family income far below the GNP per capita and have lived there for more than 20 years. The average family income of the respondent is slightly above half of the minimum standard wage according to the Indonesian Government Regulation No. 78 of 2015, which is 3,7 million rupiah (US$ 258.93) for workers in the area [19]. Therefore, based on the World Bank category for poverty, the majority of the household in the study site falls into extreme poverty [20]. The total monthly expenses of the respondent's parent is 1 million rupiah (US$ 69.98), so it is quite reasonable if the respondent's parents do not have enough options to add to other nutritional needs, such as vitamins or other nutrients besides the daily staple food.

One of the causes of high BLL in children aged 1–5 years is the habit of pica and hand to mouth, but in this study, the percentage of these two factors was found to be very low. When viewed from the history of the average length of stay of the parents of children, which is around 23 years, the high exposure to lead in the blood of children was likely obtained since they were in the mother's womb, because during pregnancy, a mother can transfer lead in her body to the fetus she is carrying. In addition, from the results of the questionnaire, it is evident that almost the entire population of Cinangka occupies an area that is entirely contaminated

**Table 3. Relationship of BLL with hematological parameters.**

| BLL | Hb Level | | | | Total | | | OR | p-value |
|---|---|---|---|---|---|---|---|---|---|
| | Anemia | | Normal | | | | | (95% CI) | |
| | n | % | n | % | n | % | | | |
| High | 30 | 33.7 | 59 | 66.3 | 89 | 100,0 | | 1.017 | 1.000 |
| Low | 13 | 33.3 | 26 | 66.7 | 39 | 100,0 | | (0.458–2.258) | |
| BLL | Basophilic Stippling | | | | Total | | | OR | p-value |
| | Yes | | No | | | | | (95% CI) | |
| | n | % | n | % | n | % | | | |
| High | 27 | 30.3 | 62 | 66.7 | 89 | 100,0 | | 1,991 | 0.214 |
| Low | 7 | 17.9 | 32 | 82.1 | 39 | 100,0 | | (0.782–5.068) | |

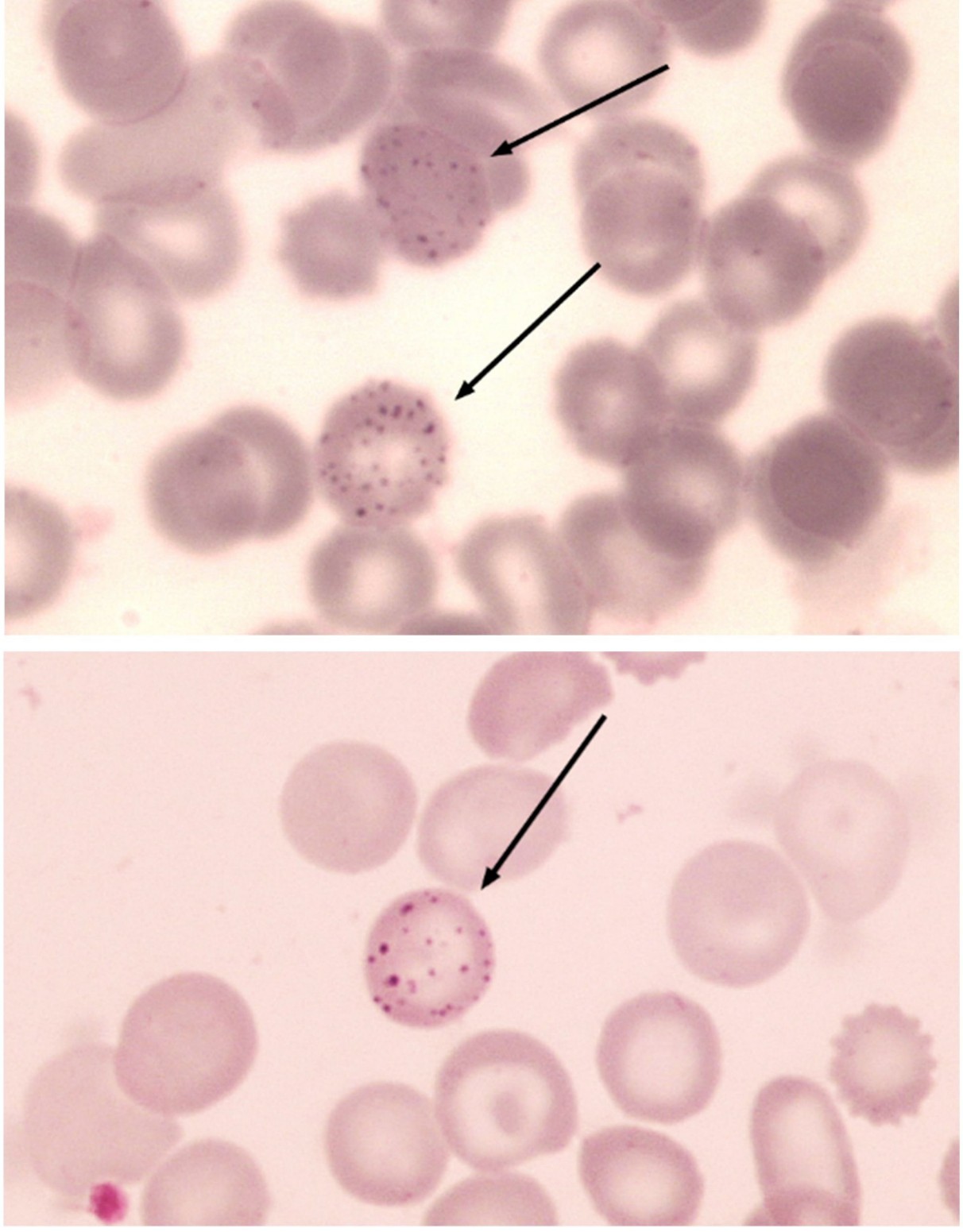

Figure 2.

**Fig 3. The appearance of basophilic stippling (objective 1.000x, Giemsa stain).**

with lead as a cause rampant smelting activities during the period of 1997–2011. Although the intensity of ULABs recycling activities continuously decrease until the formal closure of the site in October 2018, it does not automatically eliminate persistent lead exposure in the environment. Exposure to lead through drinking water was likely minimal as the Cinangka people rarely consume well water and prefer tap water provided by the government.

The educational background of the respondents' parents revealed that, most of them has a low education level and, is likely to influence their perceptions as well as attitudes towards the dangers of lead exposure to their children from the environment. Furthermore, from the aspect of parental smoking habits, it is known that only a small proportion of parents admit that they do not smoke. This fact shows that the contribution of cigarettes consumed by parents has a large enough opportunity to increase lead exposure in children because cigarettes naturally contain lead in their constituent components [21].

The BLL of the children in the surveyed area is very high based on the US CDC criteria in 1991, which is 10 μg/dL [18]. If the new BLL criteria set by the US CDC in 2012 (5 μg/dL) is applied, only 3 subjects have BLL under the safe criteria [22]. The findings indicate that the lead exposure in the study sites is still ongoing although the ULAB recycling activities have already been closed few years ago. There are several factors that may support for the high BLL of the children in the area, such as the ULAB recycling activities may still be done clandestinely, or decontamination of the environment may not be properly implemented. Nevertheless, the evidence alerts to the need for immediate termination of all ULAB recycling activities and implementation of mitigations efforts to reduce the lead exposure.

The high BLL in children living around the former ULAB area found in this study is in line with Prihartono's study, which examined 279 children aged 1–5 years in the ULABs recycling area in three border areas of Greater Jakarta. The study found that 47% of children had BLL >5 μg/dL, and 9% had BLL>10 μg/dL [23]. Other study related to lead conducted on 63 elementary school children in Cinangka Village in 2014 found that 61.2% of children had BLL > 10 μg/dL with an average BLL of 14.70 μg/dL [24]. These findings are also in line with the statement that BLL varies between countries and ethnicities, depending on location, industrialization, and lead exposure. Cinangka village with a history of ULAB recycling activities for decades, has contributed to lead contamination of the environment in the village, such as soil water and air. Ericson et al. estimated that 10,599–29,241 informal ULAB activities currently pose a risk to humans in 90 countries. The study calculated that the average BLL of children aged 0–4 years in the ULAB recycling area was 31.15 μg/dL [25]. Another study stated that 47% of children in India have an average BLL > 10 μg/dL, more than half of them live in urban and rural areas [26]. A study in Dong May Village, Vietnam, known as a ULAB recycling centre since 1980, revealed that out of 109 ten years old children who were respondents in the study, all of them had high BLL, ranging from 12 to more than 65 μg/dL [27].

Based on the children's BLL findings in Cinangka Village, it is assumed that lead accumulation has occurred in the soil around the ULABs recycling area where the residents of Cinangka Village lived. The Toxicity Characteristic Leaching Procedure (TCLP) test at one of the recycling sites and the total lead concentration analysis in soil samples in Cinangka Village on October 29, 2018, showed a lead level of 156.490 mg/L and 27,477.1 mg/kg, respectively. All those findings supported that the lead level in the Cinangka's soil was very high [28] and exceeds the standard value of soil lead contamination recommended by the Government of Indonesia, which is 3 mg/L and 6,000 mg/Kg [29]. Another study that measures the lead levels in soil at eight points in four hamlets in Cinangka Village in March 2019 revealed an average soil lead level of 4,448,213 ppm, exceeding more than ten times the US EPA (400 ppm) soil quality standard for children's play areas [30]. Based on these findings, it is believed that lead contamination is still ongoing in Cinangka Village today through the soil.

This study did not find a significant relationship between BLL and the nutritional status of the respondents. This finding is in line with study in Mongolia which did not find a relationship between BLL and children's weight or height [31]. It is also in line with a cohort study conducted on 1074 newborns in Australia [32]. Height and weight variables generally do not stand alone in terms of influencing the absorption of BLL due to other nutritional conditions, such as iron and calcium intake. Although not statistically significant, children with high BLL in this study had a greater risk of underweight and abnormal BMI. The high BLL seems to protect the children from stunting (less height for age) although the relationship was not statistically significant. In this regard, despite a clear physical change during the first 2,000 days of life, emotional, social, and intellectual performance of the children is much more difficult to measure in a cross-sectional setting.

This study also shows that there was no relationship between BLL and hemoglobin levels. This result is in line with the study of Bagaswoto et al. [33], which found a non-significant correlation between BLL and hemoglobin in 65 children aged 1–6 years in Yogyakarta and Bass's study states that hematological changes due to lead exposure only occur at high concentrations of lead exposure (>70 μg/dL) [34]. This opinion is supported by the results of a study [35] which stated that only severe acute lead poisoning was associated with hemolytic anemia. Another study that is in line with the results of this study is Anashr's study which did not find a relationship between BLL in elementary school children in Cinangka Village and hemoglobin levels [24]. One of the factors causing the absence of a relationship between BLL and anemia is the cut-off point for low BLL (10 μg/dL). The relationship between BLL and anemia, based on previous studies, was not found unless at higher lead exposures (70 μg/dL). Although this study also found two respondents whose BLL exceeded the measurement limit of the LeadCare® II instrument (>65.0 μg/dL), the data were not representative enough to be analyzed statistically.

This study found that children with high lead levels were almost twice as likely to experience inhibition of heme synthesis, as evidenced by the finding of basophilic stippling, although the relationship was not statistically significant. The finding of basophilic stippling in erythrocytes of many children in this study clearly indicates the existence of lead toxicity manifested in the form of red blood cell morphologic abnormalities which at the same time indicate interference on heme synthesis, although it has not yet caused anemia [36].

Our research provides an overview of BLL and hematological disorders in children aged 1–5 years in Cinangka Village which has never been carried out after the closure of ULAB recycling in Cinangka Village in 2018. Most anemia in children in Indonesia is classified as nutritional anemia, such as the lack of iron intake [37], for that provision of iron supplements accompanied by regular monitoring of BLL is highly recommended to mitigate the lead toxicity, especially for those who live in risk areas, such as former ULAB recycling.

In conclusion, this study provides evidence of high BLL among the young children living in the ULAB recycling site in Bogor with a sign chronic lead toxicity. Immediate action to mitigate the lead exposure as well as management of the children chronic lead toxicity is required to save the life and the future of resident living in the village and the surroundings. Subsequent study to determine the impact of chronic lead exposure on the elder (school age) children in mandatory to better evaluate the impact on IQ, behavior, and other mental disorder.

## Acknowledgments

This research is part of YI doctoral program at the Faculty of Public Health, Universitas Indonesia (FKM UI). We thank all the volunteers who participated in this study. We are also very grateful to Komite Penghapusan Bensin Bertimbal (KPBB), Eijkman Institute for Molecular

Biology, which is part of the Ministry of Research and Technology/National Agency for Research and Innovation (RISTEK-BRIN), and the Ministry of Finance's Educational Fund Management Institution, Lembaga Pengelola Dana Pendidikan, abbreviated LPDP for their support.

## Author Contributions

**Conceptualization:** Yana Irawati, Haryoto Kusnoputranto, Umar Fahmi Achmadi.

**Data curation:** Yana Irawati, Din Syafruddin.

**Formal analysis:** Yana Irawati.

**Investigation:** Yana Irawati, Alfred Sitorus.

**Methodology:** Yana Irawati, Haryoto Kusnoputranto, Umar Fahmi Achmadi.

**Project administration:** Yana Irawati.

**Resources:** Ahmad Safrudin, Alfred Sitorus, Rifqi Risandi, Suradi Wangsamuda, Puji Budi Setia Asih, Din Syafruddin.

**Supervision:** Yana Irawati, Din Syafruddin.

**Validation:** Yana Irawati, Alfred Sitorus, Suradi Wangsamuda, Din Syafruddin.

**Visualization:** Yana Irawati.

**Writing – original draft:** Yana Irawati.

**Writing – review & editing:** Puji Budi Setia Asih, Din Syafruddin.

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
