## [Decision Letter · Decision Letter 0]

16 Dec 2021

PONE-D-21-33664Blood lead levels and lead toxicity in children aged 1-5 years of Cinangka Village, Bogor RegencyPLOS ONE

Dear Dr. Irawati,

Thank you for submitting your manuscript to PLOS ONE. After careful consideration, we feel that it has merit but does not fully meet PLOS ONE’s publication criteria as it currently stands. Therefore, we invite you to submit a revised version of the manuscript that addresses the points raised during the review process.

We look forward to receiving your revised manuscript.

Kind regards,

Flavio Manoel Rodrigues Da Silva Júnior

Academic Editor

PLOS ONE

Journal Requirements:

4.Thank you for stating the following in the Funding Section of your manuscript: 

(This research funding was supported Ministry of Research and Technology/National Agency for Research and Innovation (RISTEK-BRIN), through Eijkman Institute for Molecular Biology, and Indonesia Endowment Fund for Education, abbreviated LPDP (Lembaga Pengelola Dana Pendidikan), Ministry of Finance, Indonesia.)

We note that you have provided funding information in your Funding Statement. However, funding information should not appear in the Acknowledgments section or other areas of your manuscript. We will only publish funding information present in the Funding Statement section of the online submission form. 

(This research funding was supported Ministry of Research and Technology/National Agency for Research and Innovation (RISTEK-BRIN), through Eijkman Institute for Molecular Biology, and Indonesia Endowment Fund for Education, abbreviated LPDP (Lembaga Pengelola Dana Pendidikan), Ministry of Finance, Indonesia (PRJ-26/LPDP.4/2020)

Yes. Eijkman Institute for Molecular Biology support in the field assistances, laboratory consumables and equipment facilities during the research. Also had role in data collection and analysis, decision to publish, or preparation of the manuscript.

No, LPDP had no rule in study design, data collection and analysis, decision to publish, or preparation of the manuscript.)

5. We note that Figure 1 in your submission contain map images which may be copyrighted. All PLOS content is published under the Creative Commons Attribution License (CC BY 4.0), which means that the manuscript, images, and Supporting Information files will be freely available online, and any third party is permitted to access, download, copy, distribute, and use these materials in any way, even commercially, with proper attribution. For these reasons, we cannot publish previously copyrighted maps or satellite images created using proprietary data, such as Google software (Google Maps, Street View, and Earth). For more information, see our copyright guidelines: http://journals.plos.org/plosone/s/licenses-and-copyright.

Reviewers' comments:

Reviewer's Responses to Questions

**Comments to the Author**

1. Is the manuscript technically sound, and do the data support the conclusions?

Reviewer #1: Yes

Reviewer #2: Yes

2. Has the statistical analysis been performed appropriately and rigorously? 

Reviewer #1: Yes

Reviewer #2: Yes

3. Have the authors made all data underlying the findings in their manuscript fully available?

Reviewer #1: Yes

Reviewer #2: Yes

4. Is the manuscript presented in an intelligible fashion and written in standard English?

Reviewer #1: Yes

Reviewer #2: Yes

5. Review Comments to the Author

Reviewer #1: GENERAL COMMENTS

First, congratulations on the good work! It is particularly a topic that interests me, but it is also a worrying issue that should be highlighted.

In second, the manuscript is well written and tells a reader-friendly story (context). I made some notes that I believe will further improve the quality of your work.

INTRODUCTION

- Lines 62-64: I suggest to say that this number (900,000) is almost equivalent to the number of deaths from HIV/AIDS and is greater than the number of deaths from malaria, war and terrorism and natural disasters. Saying it "is greater than" other causes of death adds more weight to your argument.

- Line 64: First occurrence of "IHME", so use the full name accompanied by the acronym. Even with a list of abbreviations, it's interesting to identify the acronym on its first appearance.

- Line 87: Same for “NHANES”.

- Line 123: Missing an L in the acronym.

MATERIALS AND METHODS

The economic issue is very present in the study. I think it would be interesting to point out, if available, general economic indicators for the region (Gross Domestic Product or Gini Coefficient, for example). In addition, informing the number of inhabitants in the region can also contribute to an understanding of the economic context of the study area.

RESULTS

- Line 166 (and elsewhere): Change g/dl to g/dL.

- Lines 166-168: Is there a limit (lower and/or higher) for hemoglobin levels? If so, it would be interesting to add them to this paragraph. For the reader, it is interesting to know the threshold used to consider the child as having anemia.

- Lines 185-186: Did you mean "50% of PARENTS lived at the ULAB recycling site for more than 23 years"?

DISCUSSION

- Lines 192-193: "Practically" and "only" together doesn't seem to make sense. Also, reiterate which “safe criteria” you are referring to.

- Lines 222-225: What is the value recommended by Management of Hazardous and Toxic Waste? It is important information to incorporate into the manuscript.

- Line 366: Apply the correct tab to the paragraph (here and elsewhere in the manuscript).

Reviewer #2: Line 104: Please include the full the names before each acronyms.

Line 138 - 140: Please, the author must describe in detail anthropometric data. Where the anthropometric measurements were taken? by whom? They are trained to ensure technique standardization? Witch technics and instruments were used to measure height and weight?

line 153: In methods section it is important to describe the currency used to measure family income. Furthermore, I suggest that the authors describe how much 2,000,000 rupiah is equal to the dollar.

Line 141 – 151: Please, in the paragraph of blood sampling and analysis, inform the reference values of all hematological parameter evaluated and how the author determine high/low levels to Hb and BLL.

Line 163 – 166: I suggest include a figure of these BLL data. The authors would include a scatter plot, highlighting the mean value of BLL and WHO reference value (10 µg/dL). This will bring greater emphasis to the outcome of the study.

263 – 264: “in the age range of 1-5 years, the growth of children does not look significant compared to when they were in elementary school.” This is not true. There are several recently studies that point out that the first 2.000 days of child’s life (5 years old) are a crucial period to development (physical, emotional, social, intellectual), than any other time. In this period children are at their developmental peak. Please, review this information in the manuscript.

Major Coments:

# According to a hierarchical model of study variables, the first level would be the sociodemographic factors, then the nutritional status and then the hematological parameters. Therefore, I suggest reverted the describe order of the methodology and results section. First the authors would describe socioeconomic characteristics, then nutritional status and finally the main outcome (hematological parameters/Hb levels). In this way, table 4 and table 1 can be unified, since socioeconomic characteristics can also be considered baseline characteristics.

# In addition to height e weigh, the author must include in Table 1 anthropometric indices as BMI for age, since the systems of classification of the nutritional status of children have been recommended by the World Health Organization and others international organization, based on the BMI for each age. The weight and height alone is of little use and effective for a child's nutritional assessment

6. PLOS authors have the option to publish the peer review history of their article (what does this mean?). If published, this will include your full peer review and any attached files.

Reviewer #1: **Yes: **Rodrigo de Lima Brum

Reviewer #2: No

---

## [Author Response · Author response to Decision Letter 0]

1 Jan 2022

We appreciate the valuable comments made by the reviewers to our manuscript. In the light of the comments, we revised the manuscript point by point. Here are our responses to reviewers ‘comments: 

Reviewer #1: 

Lines 62-64: I suggest to say that this number (900,000) is almost equivalent to the number of deaths from HIV/AIDS and is greater than the number of deaths from malaria, war and terrorism and natural disasters. Saying it "is greater than" other causes of death adds more weight to your argument.

Authors Answer: 

We do agree with the reviewer suggestion and have changed the relevant sentences in the manuscript as follows: Lead accounts for 1.5% (900,000) of deaths annually in the world, a number that is almost equivalent to the number of deaths from HIV/AIDS (954,000) and is greater than the other causes of death (lines 65-67 of the revised manuscript). 

Line 64: First occurrence of "IHME", so use the full name accompanied by the acronym. Even with a list of abbreviations, it's interesting to identify the acronym on its first appearance.

Authors Answer:

We have added to the manuscript. IHME stands for Institute for Health Metrics and Evaluation (line 67 of the revised manuscript).

Line 87: Same for “NHANES”

Authors Answer:

We have added to the manuscript accordingly. NHANES stands for National Health and Nutrition Examination Survey (line 90-91 of the revised manuscript).

Line 123: Missing an L in the acronym

Authors Answer:

We have added to the manuscript (line 127). BLL stands for Blood Lead Level. 

The economic issue is very present in the study. I think it would be interesting to point out, if available, general economic indicators for the region (Gross Domestic Product or Gini Coefficient, for example). In addition, informing the number of inhabitants in the region can also contribute to an understanding of the economic context of the study area.

Authors Answer:

We have added to the manuscript the information that the reviewer suggested, such as the number of inhabitants, Gross National Product per capita, and the minimum wage of the province. The complete statement (line 133-140) as follows: 

Cinangka Village is a rural area, located approximately 60 km from Jakarta. Its total population is 13,253, consisted of 4,195 households with the population density of 3,898 people per km2. The average income of the household is 2 million rupiah (US$ 132), which is far below the GNP per capita US$ 3, 048. Most of the resident work as subsistence farmer and ULAB recycling. Historically, this village has been used as informal battery recycling since 1978 but was temporary closed during 2003-2004, although illegal ULAB recycling activity still ongoing till now. 

Line 166 (and elsewhere): Change g/dl to g/dL

Authors Answer:

We have changed the unit from g/dl to g/dL in any relevant sentences throughout the manuscript.

Lines 166-168: Is there a limit (lower and/or higher) for hemoglobin levels? If so, it would be interesting to add them to this paragraph. For the reader, it is interesting to know the threshold used to consider the child as having anemia

Authors Answer:

We have added to the manuscript the cut off for determining the status of the haemoglobin Ievel according to WHO (see reference number 17), Hb ≥ 11 g/dL (normal) and Hb < 11 g/dL (anemia).

Lines 185-186: Did you mean "50% of PARENTS lived at the ULAB recycling site for more than 23 years"?

Authors Answer:

Yes, based on their response to the questionnaire, we found out that more than 50% of the parent of the subjects have already settled in the ULAB recycling site for more than 23 years (line 191-192 of the revised manuscript).

Lines 192-193: "Practically" and "only" together doesn't seem to make sense. Also, reiterate which “safe criteria” you are referring to.

Authors Answer:

We have re-phrased the relevant sentence accordingly. The BLL criteria was based on United States CDC. 

The BLL of the children in the surveyed areas is very high based on the US CDC criteria in 1991, which is 10 g/dL. If the new BLL criteria set by the US CDC in 2012 (5 g/dL) is applied, only 3 subjects have BLL under the safe criteria (line 248-250 of the revised manuscript). 

Lines 222-225: What is the value recommended by Management of Hazardous and Toxic Waste? It is important information to incorporate into the manuscript.

Authors Answer:

The standard value of soil lead contamination recommended by the Government of Indonesia is 3 mg/L and 6,000 mg/Kg (line 282-283 of the revised manuscript, see also reference number 29). 

Line 366: Apply the correct tab to the paragraph (here and elsewhere in the manuscript)

Authors Answer:

We have changed the format accordingly.

Reviewer #2: 

Line 104: Please include the full the names before each acronyms

Authors Answer:

We have already addressed this issue throughout the manuscript. ULAB stands for Used Lead Acid Battery (ULAB).

Line 138 - 140: Please, the author must describe in detail anthropometric data. Where the anthropometric measurements were taken? by whom? They are trained to ensure technique standardization? Witch technics and instruments were used to measure height and weight?

Authors Answer

The children’s bodyweights and body height were measured by trained village health volunteers under the supervision of nurse at Primary Health Center, using a bathroom scale (Oxone OX 917-3, Indonesia), and heights were measured using a standard mechanical stadiometer (OneMed Statue Meter, Sidoarjo, Indonesia). Body mass index was then calculated using body weight per age and height per age using the WHO Anthro program (line 159-164 of the revised manuscript).

Line 153: In methods section it is important to describe the currency used to measure family income. Furthermore, I suggest that the authors describe how much 2,000,000 rupiah is equal to the dollar

Authors Answer:

We do agree with the reviewer suggestion and the rupiah-US $ dollar conversion rate for 2,000,000 rupiah is equal to 139.28 dollar (line 150, line 188-189 of the revised manuscript).

Line 141 – 151: Please, in the paragraph of blood sampling and analysis, inform the reference values of all hematological parameter evaluated and how the author determine high/low levels to Hb and BLL.

Authors answer: 

The reference cut off value for Hb level is 11 g/dL (see reference number 17). Subjects who had Hb level below 11 g/dL is considered anemic. The reference cut off value of BLL is 10 µg/dL (see reference number 18). Subjects who had BLL value above 10 µg/dL is considered high whereas as below is normal. We determined the Hb and BLL parameter using HemoCue and LeadCare® II Portable Analyzer, as we had explained in the paraghraph (line 170-173 of the revised manuscript).

Line 163 – 166: I suggest include a figure of these BLL data. The authors would include a scatter plot, highlighting the mean value of BLL and WHO reference value (10 µg/dL). This will bring greater emphasis to the outcome of the study.

Authors Answer:

We do agree with the reviewer suggestion and have incorporated the BLL figure in the manuscript accordingly (Fig 2).

Lines 263 – 264: “in the age range of 1-5 years, the growth of children does not look significant compared to when they were in elementary school.” This is not true. There are several recently studies that point out that the first 2.000 days of child’s life (5 years old) are a crucial period to development (physical, emotional, social, intellectual), than any other time. In this period children are at their developmental peak. Please, review this information in the manuscript.

Authors Answer:

We have revised the manuscript to adopt the reviewer suggestion and have incorporated the BMI for age values based on WHO Anthro. In this regard, despite a clear physical change during the first 2,000 days of life, emotional and social and intellectual performance of the children is much more difficult to measure in a cross-sectional setting (line 296-298 of the revised manuscript). 

# According to a hierarchical model of study variables, the first level would be the sociodemographic factors, then the nutritional status and then the hematological parameters. Therefore, I suggest reverted the describe order of the methodology and results section. First the authors would describe socioeconomic characteristics, then nutritional status and finally the main outcome (hematological parameters/Hb levels). In this way, table 4 and table 1 can be unified, since socioeconomic characteristics can also be considered baseline characteristics.

Authors Answer:

We do agree with the reviewer suggestion and have changed the order of the presentation in the abstract, methodology, result and discussion accordingly.

# In addition to height e weigh, the author must include in Table 1 anthropometric indices as BMI for age, since the systems of classification of the nutritional status of children have been recommended by the World Health Organization and others international organization, based on the BMI for each age. The weight and height alone is of little use and effective for a child’s nutritional assessment

Authors Answer:

We do agree with the reviewer suggestion and have changed the relevant sentences throughout the manuscript. We have incorporated the BMI for age values in the Table 1. 

With regard to the editorial comments on the copyright issue of the figure, we have revised the figure 1 to follow the suggestion and have put the source in the Fig 1 Legend (https://www.naturalearthdata.com/) (line 480 of the revised manuscript).

---

## [Editor Report · Decision Letter 1]

7 Feb 2022

Blood lead levels and lead toxicity in children aged 1-5 years of Cinangka Village, Bogor Regency

PONE-D-21-33664R1

Dear Dr. Irawati,

We’re pleased to inform you that your manuscript has been judged scientifically suitable for publication and will be formally accepted for publication once it meets all outstanding technical requirements.

Kind regards,

Flavio Manoel Rodrigues Da Silva Júnior

Academic Editor

PLOS ONE

---

## [Editor Report · Acceptance letter]

11 Feb 2022

PONE-D-21-33664R1 

Blood lead levels and lead toxicity in children aged 1-5 years of Cinangka Village, Bogor Regency

Dear Dr. Irawati:

I'm pleased to inform you that your manuscript has been deemed suitable for publication in PLOS ONE. Congratulations! Your manuscript is now with our production department. 

Kind regards, 

on behalf of

Professor Flavio Manoel Rodrigues Da Silva Júnior 

Academic Editor

PLOS ONE